# Do the Volume-of-Fluid and the Two-Phase Euler Compete for Modeling a Spillway Aerator?

**Lourenço Sassetti Mendes** [1,2,*] , **Javier L. Lara** [1] **and Maria Teresa Viseu** [2]

1   IHCantabria—Instituto de Hidráulica Ambiental, Calle Isabel Torres 15, 39011 Santander, Spain; jav.lopez@unican.es
2   Laboratório Nacional de Engenharia Civil, Avenida do Brasil 101, 1700-066 Lisbon, Portugal; tviseu@lnec.pt
*   Correspondence: lmendes@lnec.pt

**Abstract:** Spillway design is key to the effective and safe operation of dams. Typically, the flow is characterized by high velocity, high levels of turbulence, and aeration. In the last two decades, advances in computational fluid dynamics (CFD) made available several numerical tools to aid hydraulic structures engineers. The most frequent approach is to solve the Reynolds-averaged Navier–Stokes equations using an Euler type model combined with the *volume-of-fluid* (*VoF*) method. Regardless of a few applications, the *complete two-phase Euler* is still considered to demand exorbitant computational resources. An assessment is performed in a spillway offset aerator, comparing the *two-phase volume-of-fluid* (*TPVoF*) with the *complete two-phase Euler* (*CTPE*). Both models are included in the OpenFOAM® toolbox. As expected, the *TPVoF* results depend highly on the mesh, not showing convergence in the maximum chute bottom pressure and the lower-nappe aeration, tending to null aeration as resolution increases. The *CTPE* combined with the *k–ω SST Sato* turbulence model exhibits the most accurate results and mesh convergence in the lower-nappe aeration. Surprisingly, intermediate mesh resolutions are sufficient to surpass the *TPVoF* performance with reasonable calculation efforts. Moreover, compressibility, flow bulking, and several entrained air effects in the flow are comprehended. Despite not reproducing all aspects of the flow with acceptable accuracy, the *complete two-phase Euler* demonstrated an efficient cost-benefit performance and high value in spillway aerated flows. Nonetheless, further developments are expected to enhance the efficiency and stability of this model.

**Keywords:** spillway aerator; aeration; CFD; two-phase Euler; volume-of-fluid; hydraulic structures

## 1. Introduction

Spillways are essential for dam safety, controlling the reservoir water level and discharge, and preventing dam overtopping, one of the leading causes of structural failure and rupture. Such a crucial hydraulic structure requires careful design validation. Typically, a spillway is constituted by: intake, weir, chute, energy dissipation structure, and river restitution. The water flow from the reservoir approaches the intake in the subcritical regime with low velocity. Next, the flow is guided into the weir, the control section, where the critical regime is attained. Downstream, the chute flow is supercritical, with high velocities (frequently larger than 20 m/s), high levels of turbulence, and significant air entrainment and transport. Frequently, the geometry imposes complex flow patterns, such as crosswaves, significant free-surface variations, and lateral-mixing. At the end of the spillway, a structure dissipates energy, allowing proper restitution to the river stream. Moreover, surrounding and entrained air plays a crucial role in the safe operation of spillways [1] and may significantly alter the flow characteristics. Aeration must be considered due to the effects of flow bulking, drag reduction, prevention or mitigation of cavitation damage, reduction of the breakup length of water jets discharging into atmosphere, interaction with the turbulence field, re-oxygenation, and transference of atmospheric gases that have a vital function on stream ecology [2,3]. Besides free-surface aeration–the natural air entrainment

process along with the air-water interface of high-velocity flows [4]–aerator devices are commonly designed in chutes, constituting an economical solution to eliminate cavitation damages [5]. Due to the complexity of air-water flows, they continue to be analyzed mainly on physical models. Nevertheless, important scale-effects limitations are present, especially in aeration-related processes [6].

In the XX century, physical modeling was the reference procedure and still is nowadays. However, over the last 20 years, numerical modeling of hydraulic structures proliferated among researchers and practical engineers due to advances in hardware and numerical methods, and the availability of user-friendly computational fluid dynamics (CFD) software. Also, the scientific community published plenty of data of physical models, allowing the calibration and validation of numerical models (e.g., [7–14]). Despite not being completely established in practical engineering, CFD is becoming a reliable and important tool, enabling the test of innovative designs quickly with cost-saving, and a detailed analysis of most phenomena in the flows (e.g., [15–24]). CFD also provides valuable information in the design and validation stages, predicting problems. Moreover, CFD supplies beneficial information to build the physical model and locate measurement devices. The most promising approach is an integrated physical and numerical modeling–the hybrid modeling technique–where both models feed each other iteratively, mitigating time, costs, and limitations. Nevertheless, the most recent and complex numerical tools need proper evaluation to be introduced in engineering practice, to ensure accurate predictions to assist engineering designs.

Chanson [3] remarks that "the modeling of aerated flows is presently restricted by the complexity of theoretical equations, some limitations of numerical techniques, a lack of full-scale prototype data, and very-limited detailed experimental data sets suitable for sound CFD model validation". The absence of detailed turbulence data in most research is a major drawback.

Numerical modeling of two-phase air-water highly turbulent flows is a very challenging field, especially if aeration occurs. Regarding spillways, many difficulties arise. The large dimensions of the structures require huge computational domains that must cope with the extended range of flow characteristic lengths. The extreme velocity generates high turbulence, which is impossible to model directly at all time and length scales. Furthermore, aeration demands more complex CFD models that comply with air, water, and the mixture. After entrainment, besides being advected by the flow, air packets of different sizes (e.g., pockets, droplets, bubbles) may suffer extremely complicated processes such as fragmentation and coalescence, diffusion, dissolution, and buoyant degassing [25,26], which should all be modeled. The air-water interface is hard to determine [27,28]. Furthermore, the coupling of equations at the air-water interfaces and the turbulence interactions of the phases are extremely complex. The range of length scales is hugely extent, i.e., Kolmogorov length, bubble diameter, surface roughness, flow turbulence eddies, and flow large characteristic lengths–channels' depth or width–[3,29]. In the same way, the time scales of the acoustic and quiescent bubble phases vary from milliseconds to seconds [26].

The constant technological advances keep promoting new numerical methods and more complex models to simulate spillway flows. Several approaches are presented next. Direct Numerical Simulations (DNS) are impossible due to the extreme spatial and time resolutions required to contemplate all the processes. Complete Lagrangian models are also non-practical due to the exorbitant number of particles necessary to represent all bubbles, air, and water, resulting in a too-large computational time [27]. On the combined Euler-Lagrange method, the water phase is continuous and solved in an Euler referential. The air is represented by a dispersed phase of particles calculated by a Lagrangian approach. The number of bubbles is unreasonable for most applications, and the air volume fraction should not exceed 15% [30].

The most popular approaches to solve the Reynolds-averaged Navier–Stokes equations are the interfacial Euler models *volume-of-fluid* (*VoF*) [31] and Level–Set (LS), Ref. [32] which are meant for two immiscible fluids. Both use only a volume fraction function propagated by

an advection equation and one set of momentum equations calculated for averaged mixture flow properties. However, they are still effective tools due to the accurate free surface tracking, simplicity, stability, and low to average computational costs. The main drawbacks are surface tension calculation [33], the potential for some unrealistic cavitation and the determination of interphase surface interactions [30]. However, despite these features, they were applied to spillways with success (e.g., [19,34–36]).

Interfacial models can be combined with a sub-grid air-bubble density equation model (SGB), which depends on the local average surface-flow properties such as turbulence. The *VoF* or LS simulate the continuous water and surrounding air and the SGB the entrained air-bubble. The most critical aspects are the lack of a comprehensive air entrainment model that predicts surface aeration and degassing for distinct types of flows, the bubble transformations, and the coupling models between the continuous and dispersed fluids. Nonetheless, this method has a wide range of applications [18,26,27,37].

Mixture models allow the interpenetration of two or more fluids, and may comprehend interfacial methods between some fluids. For example, two immiscible fluids represent the continuous air and water, and a third fluid simulates the air bubble that may interpenetrate both. Therefore, interfacial boundary conditions are highly complex. Several continuity equations and only a single set of momentum equations for the averaged mixture properties are solved [30]. Several applications to spillways are found [24,38–40].

Despite the efficiency of the previous methods, the application of models based on a single set of momentum equations has generally been limited to air-water flows with small dispersed air-concentrations (<15 to 20%) as per [3,29].However, recent applications of the *VoF* and mixture models succeeded in simulating larger air concentrations, including in chute spillways [18,24]. The *complete two-phase Euler* is the most complex. Each fluid is represented by a continuous phase that interpenetrates the other and has its momentum, energy, and continuity equations. Hence, there are no limitations for the volume fraction occupied by each fluid. Nevertheless, the closure of these equations, i.e., the interphase interactions (e.g., heat and species transfer, drag, lift, turbulent dispersion), is extremely complicated and can be solved by numerous methods based on empirical relations, which tend to have limitations in scope and accuracy. The *complete two-phase Euler* is standard in nuclear and chemical engineering. However, despite the technological advances and efforts to develop new methods to overcome the vulnerabilities, further applications in hydraulic structures engineering have been discouraged due to the high complexity, convergence problems, and calculation esources. Notwithstanding, it is the better-suited multiphase model for highly aerated flows (air-concentration > 20%) [30]. A few applications evidence the importance of this tool for spillway aerated flows (e.g., [20,41,42]).

Comparisons of the previous models' performance are found in [22,43]. In summary, the most popular approaches to spillway numerical modeling suffer from significant drawbacks: lack of flow bulking in the interfacial models, and the existence of a threshold for the maximum air-concentration in the mixture and sub-grid bubble models. Despite being considered to demand exorbitant resources and be extremely difficult to use, the *complete two-phase Euler* model is conceived to simulate highly aerated flows, potentially overcoming the limitations of the previous models.

The present work compares the efficiency of the *complete two-phase Euler* model with the *VoF*. A spillway with an offset aerator device is used to evaluate both models' compliance in different 3D mesh resolutions and turbulence models. Also, calculation time and stability are analyzed. An accurate application of the *VoF* needs a fine mesh to reproduce each air bubble, which is unfeasible in CFD of hydraulic structures due to the large dimensions and the high velocity. Despite the high dependency on the mesh resolution to simulate air-entrainment, evaluating the *VoF* is important because it is still a widely used tool in practical engineering.

An offset aerator separates the spillway flow from the bottom (Figure 1), creating a jet and inducing air-entrainment through the lower surface, hereinafter named as lower-nappe aeration. Immediately downstream of the offset aerator, there is a cavity zone filled with

air, followed by the impact zone, i.e., where the water jet re-joins the bottom. The high turbulence levels cause lower-nappe aeration at the air-water interface of the cavity zone.

An unprecedented 3D numerical study is conducted due to the mesh size, the assessment of multiple flow characteristics [44], the physical model with detailed data and high Weber and Reynolds numbers that reduce scale effects [6], and the enormous computational efforts.

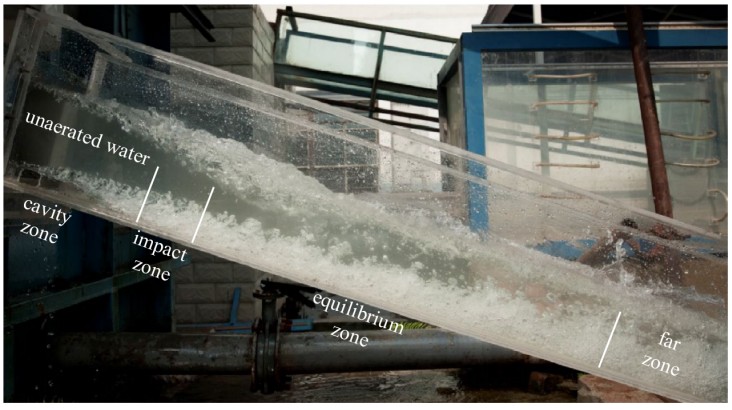

(**a**)

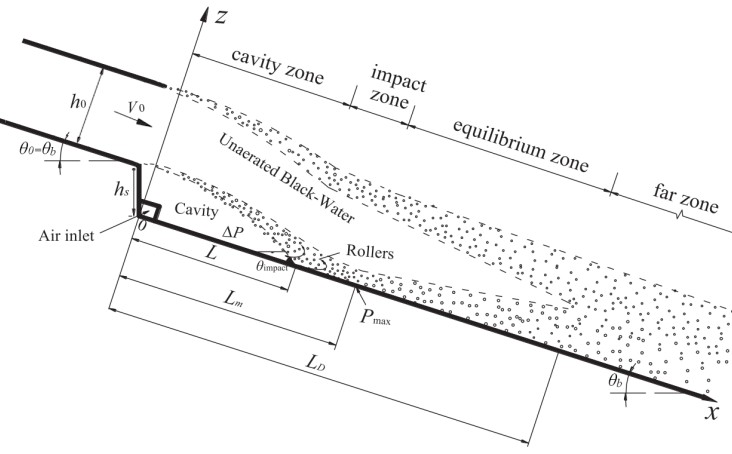

(**b**)

**Figure 1.** Laboratory setup ([45]-authorized reproduction). (**a**) Physical model (unknown flow conditions). (**b**) Flow characteristics.

The paper is organized as follows. After this introduction, the methodology is described, including the laboratory data, the mathematical models, and the numerical setup. Next, the results are analyzed. Finally, the discussion and conclusions are presented.

## 2. Methodology

The *complete two-phase Euler* and the *volume-of-fluid* models are evaluated against the physical modeling of an offset chute aerator by Bai et al. [11] (Figure 1).

The main flow characteristic to be evaluated is the lower-nappe aeration induced by the offset aerator. This phenomenon is quantified by the flow-rate of lower-nappe entrained air. The bubble diffusion is also assessed through selected water fraction profiles downstream of the impact zone.

Additionally, two characteristics related to the pressure and flow geometry downstream of the aerator are analyzed: the value and location of the maximum pressure increment at the spillway bottom and the length of the cavity zone.

A mesh dependence analysis is performed for six different resolutions. The two most popular Reynolds Averaged Navier-Stokes (RANS) turbulence models are tested in each model: $k$–$\omega$ SST and $k$–$\epsilon$. Hence, the models' performance is analyzed for a total of 24 combinations (2 models $\times$ 6 resolutions $\times$ 2 turbulence models).

### 2.1. Laboratory Data

Bai et al. [11] conducted unmatched physical modeling of a large spillway offset aerator with high velocity (4 to 9 m s$^{-1}$) and high inflow water depth (0.15 m). Hence, presenting high values of the Reynolds ($5.5 \times 10^5 < Re < 1.2 \times 10^6$) and Weber numbers ($180 < We^{0.5} < 405$) which comply with Pfister [6] criteria to mitigate aeration related scale effects: $Re > 2.2 \times 10^5$ and $We^{0.5} > 140$. Moreover, the extent number of tests and a complete set of measured flow properties (i.e., pressure at the bottom, velocity profiles, air-concentration, turbulence, and bubble sizes) provide relevant data to validate numerical models.

The rectangular channel is 5 m long and 0.25 m wide (Figure 1a). The aerator offset height ($h_s$) varies from 0.025 m to 0.045 m. The upstream emergence angle ($\theta_0$) inclination is variable from 0° to 14.1° and the channel bottom angle ($\theta_b$) ranges from 5.7° to 14.1°. The channel bed and side-walls are made of smooth polyethylene with a roughness height of $1 \times 10^{-5}$ m.

A case with inlet velocity ($V_0$) of 9 m s$^{-1}$, water depth ($h_0$) of 0.15 m, equal emergence and bottom angle of 14.1° and offset height of 0.045 m are selected to mitigate the scale effects. Froude number ($Fr$) is 7.4, $Re = 1.2 \times 10^6$ and $We^{0.5} = 405$ .

### 2.2. Mathematical Models

The CFD solvers and the turbulence models used are included in the OpenFOAM® toolbox version 2012 [46]. The *volume-of-fluid* solver is named *interFoam*. The *complete two-phase Euler* solver is named *twoPhaseEulerFoam*.

#### 2.2.1. Two-Phase Volume-of-Fluid

The *two-phase volume-of-fluid* solves the Reynolds-averaged Navier–Stokes equations for two incompressible, isothermal, and immiscible fluids. The interface capture is based on a *volume-of-fluid* (*VoF*) method that incorporates an interfacial compression flux term [47,48]. Hereinafter, this model is mentioned as *TPVoF*.

The mass Equation (1) and a single-set of momentum conservation equations (2) are solved. Hence, the two fluids—water and air—share the same velocity field.

$$\nabla \cdot \vec{V} = 0 \tag{1}$$

$$\frac{\partial \rho \vec{V}}{\partial t} + \nabla \cdot \left( \rho \vec{V} \vec{V} \right) = -\nabla p^* - \vec{g}\vec{X} \cdot \nabla \rho + \nabla \cdot (2(\mu + \mu_t)S) + \vec{F}_\sigma \tag{2}$$

$$\vec{F}_\sigma = \sigma \vec{\kappa} \nabla \alpha \tag{3}$$

where $\vec{V}$ is the RANS velocity vector, $\rho$ is the density, $t$ is the time, $p^*$ is the pseudo-dynamic pressure, $\vec{g}$ is the gravitational acceleration vector, $\vec{X}$ is the position vector, $\mu$ is the molecular dynamic viscosity, $\mu_t$ is the eddy viscosity coefficient (i.e., turbulent dynamic viscosity) and $S$ is the strain rate tensor. $\vec{F}_\sigma$ is the surface tension force term for the momentum Equation (3). $\sigma$ is the surface tension, $\vec{\kappa}$ is the curvature of the interface and $\alpha$ is the water volume fraction.

Any *VoF* phase property $\Phi$ (e.g., $\rho$, $\mu$) is a volume-average of the intrinsic fluid property of water (subscript $w$) and air (subscript $a$) (4).

$$\Phi = \alpha \Phi_w + (1 - \alpha)\Phi_a \tag{4}$$

Additionally, according to the *VoF* method implementation in OpenFOAM®, the fluids volume fraction in each cell is defined by a scalar function ($\alpha$) ranging from 0 to 1 that allows the interface tracking: $\alpha = 1$ is a water cell and $\alpha = 0$ is an air cell. Other $\alpha$

values identify interface cells. The phase advection Equation (5) comprehends an artificial compression meant to preserve a sharp interface (third term).

$$\frac{\partial \alpha}{\partial t} + \nabla \cdot \vec{V}\alpha + \nabla \cdot \vec{V}_{ic}\alpha(1 - \alpha) = 0 \tag{5}$$

$$\vec{V}_{ic} = C_\alpha |\vec{V}| \frac{\nabla \alpha}{|\nabla \alpha|} \tag{6}$$

A compression velocity ($\vec{V}_{ic}$) (6) proportional to the local velocity field magnitude is applied perpendicular to the interface. The interface compression is triggered by the $C_\alpha$ coefficient that usually ranges from 0 to 1.

The solver uses a segregated solution approach. Each time step starts an update of the interface, followed by a prediction of the velocity. A Pressure Implicit with Splitting Operators (PISO) type algorithm corrects velocity and implicitly the pressure, advancing both in time. Finally, the turbulence is calculated.

2.2.2. Complete Two-Phase Euler

The *complete two-phase Euler* solves the Reynolds-averaged Navier–Stokes equations for two interpenetrating and compressible fluid phases—a continuous (*c*) and a dispersed (*d*)—including heat transfer [49,50]. Hereinafter, this model is mentioned as *CTPE*.

The continuity equation is solved for the dispersed phase (7), considering $\alpha_c = 1 - \alpha_d$.

$$\frac{\partial \alpha_d}{\partial t} + \nabla \cdot \left( \alpha_d \alpha_c \vec{V}_r \right) + \nabla \cdot \left( \alpha_d \vec{V} \right) = \alpha_d \nabla \cdot \vec{V} + \alpha_c \nabla \cdot \vec{V}_d - \alpha_d \nabla \cdot \vec{V}_c \tag{7}$$

$$\vec{V}_r = \left( \vec{V}_d - \vec{V}_c \right) + T_{D,d} + T_{D,c} \tag{8}$$

where $\vec{V}_r$ is the phases relative RANS velocity vector, $T_{D,d}$ is the dispersed phase turbulent dispersion term and $T_{D,c}$ is the continuous phase turbulent dispersion term.

The momentum Equation (9)—written for a generic phase *i*— is solved for each phase but not directly. Instead, following Issa [51] methodology, the face flux is initially predicted and afterward corrected by the pressure, which is shared between the two phases, and solved iteratively.

$$\frac{\partial \alpha_i \rho_i \vec{V}_i}{\partial t} + \nabla \cdot \left( \alpha_{i,f} \rho_{i,f} \vec{\phi}_i \vec{V}_i \right) - \varepsilon_{cont,i} \vec{V} + (\alpha_i \rho_i + C_{vm}) \frac{\partial \vec{V}_i}{\partial t} =$$
$$\nabla \cdot R^{eff} - C_d \vec{V} - C_{vm} \left( \frac{D\vec{V}_i}{Dt} - \frac{D\vec{V}_j}{Dt} \right) \tag{9}$$

$$R^{eff}_i = \left[ \nabla \cdot \left( \alpha_i \rho_i \nu_{eff} \vec{V}_i \right) - \left( \alpha_i \rho_i \nu_{eff} \left( \nabla \vec{V}_i^T - \frac{2}{3} tr \left( \nabla \cdot \vec{V}_i \right) I \right) \right) + \alpha_i \rho_i \frac{2}{3} k_i I \right] \tag{10}$$

where $\vec{\phi}$ is the face-flux vector, $\varepsilon_{cont}$ is the continuity error, $C_d$ is the drag coeficient and $C_{vm}$ is the virtual mass coeficient. $R^{eff}$ is the stress rate tensor (10), $\nu_{eff} = \nu + \nu_t$ is the effective viscosity and $k$ is the turbulent kinetic energy.

The energy conservation equation is solved for each phase (see Equation (11)) in the form of internal energy or enthalpy. Therefore variable *he* is a hybrid variable that represents one or the other.

$$\frac{\partial \alpha_i \rho_i he_i}{\partial t} + \nabla \cdot \left( \alpha_{i,f} \rho_{i,f} \vec{\phi}_i he_i \right) + \frac{\partial \alpha_i \rho_i K_i}{\partial t} + \nabla \cdot \left( \alpha_{i,f} \rho_{i,f} \vec{\phi}_i K_i \right) - \varepsilon_{cont,i} he_i$$
$$-\varepsilon_{cont,i} K_i - \nabla^2 \left( \alpha_i \alpha^{eff}_i he_i \right) =$$
$$k_{h,i} \left( T^{IF} - T_i \right) + \left( \frac{k^{eff}_{h,i}}{C_{pv,i}} he_i^{IF} \right)^{explicit} - \left( \frac{k^{eff}_{h,i}}{C_{pv,i}} he_i \right)^{implicit} + \alpha_i \rho_i \vec{V}_i \cdot \vec{g}$$
(11)

where $K$ is the kinetic energy, $T$ is the fluid temperature, $\alpha^{eff}$ is the effective thermal diffusivity, $k_h$ is the convective heat transfer coefficient, $k^{eff}_h$ is the effective volumetric convective heat transfer coefficient and $C_{pv}$ is the specific heat capacity. The *IF* superscript refers to an interfacial property.

Several closures for heat transfer, drag, lift, and turbulent dispersion are available. Only a single-size air bubble can be adopted. The *CTPE* allows considering both fluids as continuous or dispersed, depending on the respective volume-fraction value. Using a blended interfacial model, for each cell, the solver identifies one of the following scenarios: phase 1 is dispersed and phase 2 is continuous, phase 2 is dispersed and phase 1 is continuous, or it is a cell with no obvious dispersed phase. Several blending options are available: constant, linear, and hyperbolic.

The turbulent dispersion force is defined by Burns et al. [52] and Otromke [53], as follows:

$$M_D = -C_D \frac{3}{4} \frac{\alpha_d \, \rho_c \, \nu_t}{d_d \sigma_\alpha} |\vec{V}_r| \left( \frac{1}{\alpha_d} + \frac{1}{\alpha_c} \right) \nabla \alpha_d$$
(12)

where $C_D$ is the non-dimensional drag coefficient and $\sigma_\alpha$ is the turbulent Prandtl number for interfacial area density. The turbulent dispersion force is a function of the blending interfacial model chosen and its settings. The solver calculates the turbulent dispersion at each cell for one of the previous three blending model scenarios. Hence, the turbulence dispersion action depends totally on the blending model's chosen parameters, especially on the maximum volume-fraction value to a phase be considered as dispersed. For example, the user may define this value as 0.6, i.e., if the volume-fraction values of a phase range from 0 to 0.6, that phase is considered dispersed. Higher values, consider the phase continuous or with no obvious dispersed phase, depending on the model applied.

Each time step starts by solving the phase continuity, followed by discretization and linearization of the momentum equations that predict the flux. Next, energy conservation is solved. At this moment, the pressure is solved, and the flux and velocity are corrected. Finally, the turbulence is calculated.

### 2.2.3. Turbulence Models

The $k$–$\epsilon$ and $k$–$\omega$ *SST* RANS turbulence models are among the most used in hydraulic structures, hence are applied in the numerical study of the spillway bottom aerator. The $k$–$\epsilon$ model is conceived for internal flows and is the most common in RANS simulations. The $k$–$\omega$ *SST* is used due to its advantage for boundary flows. Both models have two transport equations: one for the turbulent kinetic energy ($k$) and another for the turbulent dissipation rate ($\epsilon$) or the turbulent specific dissipation rate ($\omega$), which are used to determine the eddy viscosity. In the *volume-of-fluid*, both models do not include density explicitly. Therefore, instead of the dynamic form ($\mu_t$), is calculated the kinematic eddy viscosity (15) [54,55]. The default coefficients are employed.

The $k$–$\epsilon$ model is based on Launder and Spalding [56] and El Tahry [57], and is defined by:

$$\frac{D\rho k}{Dt} = \nabla \cdot (\rho D_{k1} \nabla k) + G_k - \frac{2}{3}\rho \left( \nabla \cdot \vec{V} \right) k - \rho \epsilon + S_k$$
(13)

$$\frac{D\rho \epsilon}{Dt} = \nabla \cdot (\rho D_\epsilon \nabla \epsilon) + \frac{C_1 G_k \, \epsilon}{k} - \left( \frac{2}{3}C_1 - C_{3,RDT} \right) \rho \left( \nabla \cdot \vec{V} \right) \epsilon - C_2 \rho \frac{\epsilon^2}{k} + S_\epsilon$$
(14)

$$\nu_t = C_\mu k^2 / \epsilon \tag{15}$$

where $D_{k1} = \nu + \nu_t/\sigma_k$ and $D_\epsilon = \nu + \nu_t/\sigma_\epsilon$ are the effective viscosity, $\nu$ is the kinematic viscosity, $G_k$ is the $k$ production rate, $S_k$ and $S_\epsilon$ are source terms. $\sigma_k = 1.0$, $\sigma_\epsilon = 1.3$, $C_1 = 1.44$, $C_{3,RDT} = 0$ and $C_2 = 1.92$ and $C_\mu = 0.09$ are the default coefficients.

The $k$–$\omega$ $SST$ turbulence model is based on Menter and Esch [58] with the contributions of Hellsten [59], Menter et al. [60], Spalart and Rumsey [61], and is defined as follows:

$$\frac{D\rho k}{Dt} = \nabla \cdot (\rho D_{k2} \nabla k) + \rho P_k - \frac{2}{3}\rho k \left(\nabla \cdot \vec{V}\right) - \rho \beta^* \omega k + S_k \tag{16}$$

$$\frac{D\rho \omega}{Dt} = \nabla \cdot (\rho D_\omega \nabla \omega) + \frac{\rho \gamma P_\omega}{\nu} - \frac{2}{3}\rho \gamma \omega \left(\nabla \cdot \vec{V}\right) - \rho \beta \omega^2 + 2\rho(1 - F_1)a_{\omega 2}\frac{\nabla k \cdot \nabla \omega}{\omega} + S_\omega \tag{17}$$

$$\nu_t = a_1 \frac{k}{max(a_1\omega, b_1 F_{23}S)} \tag{18}$$

where $D_{k2} = \nu + a_k\nu_t$ and $D_\omega = \nu + a_\omega\nu_t$ are the effective viscosity. $P_k$ and $P_\omega$ are production terms. $S_k$ and $S_\omega$ are source terms. $\beta^* = 0.09$, $a_1 = 0.31$, $b_1 = 1.0$, $c_1 = 10.0$, $F_{23}$ are coefficients. $a_k$, $a_\omega$, $a_{\omega 2}$, $\beta$, and $\gamma$ blend inner and outer coefficient values using the $F_1$ coefficient.

In the *complete two-phase Euler*, instead of the standard $k$–$\epsilon$ model, is applied the $k$–$\epsilon$ *mixture* which is a specific turbulence model for two-phase gas-liquid systems, based on Behzadi et al. [62] and Lahey [63]. This model solves a single set of equations for the air-water mixture—$m$ subscript—to determine $k_m$ (20) and $\epsilon_m$ (21). The mixture variables are calculated through an effective density weighted average of air and water properties (19).

$$\Phi_m(\Phi_w, \Phi_a) = \left(\alpha\rho_w\Phi_w + (1 - \alpha)\rho_{a,eff}\Phi_a\right)/\rho_m \tag{19}$$

$$\frac{Dk_m}{Dt} = \nabla \cdot \left(\frac{\nu_m}{\sigma_k}\nabla k_m\right) + G_m - \frac{2}{3}\left(\nabla \cdot \vec{V}_m\right)k_m - \epsilon_m + \frac{G_b k_m}{\rho_m} \tag{20}$$

$$\frac{D\epsilon_m}{Dt} = \nabla \cdot \left(\frac{\nu_m}{\sigma_\epsilon}\nabla \epsilon_m\right) + C_1 G_m \frac{\epsilon_m}{k_m} - \frac{2}{3}C_1\left(\nabla \cdot \vec{V}_m\right)\epsilon_m - C_2\frac{\epsilon_m^2}{k_m} + \frac{C_3\epsilon_m^2 G_b}{\rho_m k_m} \tag{21}$$

$$\rho_m = \alpha\rho_w + (1 - \alpha)\rho_{a,eff} \tag{22}$$

where $\rho_{a,eff} = \rho_a + C_{vm}\rho_w$. $\nu_m = \Phi_m(\nu_{t,w}, \nu_{t,a})$ is the mixture turbulent kinematic viscosity, $\nu_{t,w}$ and $\nu_{t,a}$ are water and air tubulent kinematic viscosity. $G_m$ is the $k_m$ production rate, $\vec{V}_m$ is the mixture velocity vector, $G_b$ is the $k_m$ production rate by air-bubble, $\rho_w$ is the water density, $\rho_a$ is the air density and $\rho_m$ is the mixture density (22). $C_{vm}$ is the virtual mass coeficient. $C_3 = 1.92$. The remaining constants share same values with the $k$–$\epsilon$ model. Furthermore, the $k$–$\epsilon$ *mixture* model is improved following Weller et al. [64]. Thus, a phase fraction limiter is applied to the bubble-generated turbulence ($G_b$), avoiding spurious turbulence generation where bubbles are not present. In the current work, $G_b$ is only activated if $\alpha_d > 0.3$, a standard value.

In the *complete two-phase Euler*, the standard $k$–$\omega$ $SST$ model is modified to include the bubble-induced turbulent viscosity model of Sato et al. [65]. Thus, a term is added to the turbulent viscosity Equation (18), as follows:

$$\nu_t^{Sato} = a_1 \frac{k}{max(a_1\omega, b_1 F_{23}S)} + \left(1 - e^{-y^+/16}\right)^2 c_b d_b \alpha_d \vec{V}_r \tag{23}$$

where $y^+$ is the dimensionless distance from the wall, $c_b = 0.6$ and $d_b$ is the characteristic bubble size. Hereinafter, this model is mentioned as $k$–$\omega$ $SST$ *Sato*.

### 2.3. Numerical Setup

The numerical domain (Figure 2) is defined by a nozzle with 0.5 m of length and a rectangular section with 0.25 m of width per 0.15 m of height ($h_0$), followed by a rectangular

channel with 2 m of length, 0.5 m of height, and the exact width of the nozzle. On each side wall, by the bottom and next to the step, is placed an air-vent with $0.02 \times 0.02$ m$^2$. The nozzle and channel bottom have a downward slope angle of 14.1°.

Rules of thumb in hydraulic structures CFD recommend a minimum of 10 to 20 cells per characteristic hydraulic length (e.g., water depth, pipe radius, etc.). Nevertheless, the dependence of the result on the mesh resolution must be analyzed for every flow and numerical setup. Thus, a set of fully orthogonal and hexahedral 3D meshes with 6 different resolutions ($R_m$) relative to the nozzle height ($h_0$) is considered (Table 1). The cell edge length ($d_m$) ranges from 2.5 mm ($R_m = 60$) to 15 mm ($R_m = 10$). Mesh resolution is limited to 60 cells due to the extremely small time-step needed to verify the Courant–Friedrichs–Lewy condition ($\Delta t < d_m/V_0 = 2.5 \times 10^{-3}/9 = 2.8 \times 10^{-4}$ s) that leads to impractical calculation times, even in high-performance computing clusters (HPC). To calculate in parallel at the HPC, the mesh was decomposed into 16 to 256 sub-domains, using the Scotch method.

**Table 1.** Mesh.

| $R_m$ | Edge Length [mm] | Total Cells |
|---|---|---|
| 10 | $h_0/10 \approx 15$ | 85,510 |
| 15 | $h_0/15 \approx 10$ | 273,750 |
| 20 | $h_0/20 \approx 7.5$ | 684,080 |
| 30 | $h_0/30 \approx 5$ | 2,190,000 |
| 40 | $h_0/40 \approx 3.75$ | 5,472,640 |
| 60 | $h_0/60 \approx 2.5$ | 17,520,000 |

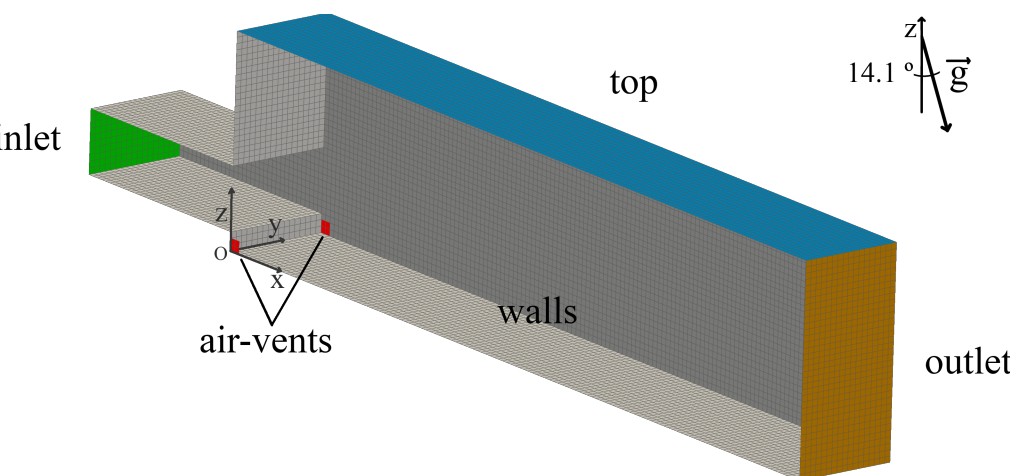

**Figure 2.** Mesh-boundary.

The boundary conditions are presented next. At the inlet, the velocity ($V_0$) is 9 m s$^{-1}$; the pressure is automatically calculated to assure the flux. The characteristic turbulent mixing length is 0.0105 m ($0.07h_0$). At the top, outlet and air vents, a total pressure condition is defined. At the outlet, the velocity condition is zero gradient for outflow, and the inflow is blocked. Both top and air vents have a binary velocity condition: zero gradient for outflow and exclusive pressure-driven normal air inflow. Turbulence for the previous three boundaries is zero gradient for outflow and for inflow $k = 3.1 \times 10^{-5}$ m$^2$ s$^{-2}$, $\epsilon = 1.6 \times 10^{-6}$ m$^2$ s$^{-3}$ and $\omega = 0.58$ s$^{-1}$, considering a turbulent mixing length equal to the channel width (0.25 m) and an intermediate turbulent intensity of 0.05. Walls have no-slip tangent velocity and a turbulent wall function for the roughness of $1 \times 10^{-5}$ m, as described in the physical modeling. Due to the absence of specification of the inflows' turbulence intensity (TI) in the physical modeling, two values were tested: 0.005 and 0.2. In

the *complete two-phase Euler*, the temperature of the fluids is intended to be 300 K (26.85 °C). Hence, the initial temperature of the domain and the inflow temperature are set to 300 K, and the walls have a zero-gradient condition.

An arbitrary bubble diameter of 0.1 mm is adopted after observing the multitude of sizes of air-bubbles and air-packets in Figure 1a). Furthermore, to respect the *CTPE*'s conception, the bubble diameter must be inferior to the cells' size.

The following numerical settings are used: Euler time derivative; gradient is Gauss linear with multi-directional cell limiter 1.0; the divergence scheme is linear-upwind for velocity. For turbulence fields, i.e., $k$, $\epsilon$ and $\omega$, the divergence scheme is upwind. Regarding the *TPVoF* model, interface compression is based on a generic limited scheme for the divergence of $\alpha$ and an interface compression coefficient ($C_\alpha$) of 0.5. For the *CTPE* model, among the available closures, the following models were used: Schiller and Naumaan [66] for drag, Ranz and Marshall [67] for heat transfer, Tomiyama et al. [68] for lift and the turbulent dispersion method based on Burns et al. [52] and Otromke [53]. This study applies a hyperbolic blending function that considers the fluid continuous if the phase volume-fraction value is larger than 0.6.

The simulations are transient. The PISO pressure-velocity algorithm is used with standard settings. Hence data is sampled every time step (approximately 2000 to 10,000 Hz, depending on the mesh) during 1 s and averaged after a warm-up period where semi-steady flow conditions are reached. Approximate time-independence is attained for lower-nappe aeration ($\beta$), maximum pressure at chute bottom ($\Delta p^b_{max}$), open-boundaries air and water flow-rate, and several other domain properties: water and air volume, total kinetic energy, total turbulent kinetic energy and minimum, and maximum pressure. The presented profile plots and bottom data are collected in the central plane along the chute, i.e., the mid-plane of the chute.

Calculations were performed in two HPC clusters composed of Intel Xeon E5-2680 or AMD EPYC 7501 CPUs (Central Process Unit). Simulations ranged from 16 to 256 cores and needed 10 to 50,000 CPU.core hours (2000 CPU core.days). Due to the extremely high calculation time, the two highest resolution meshes ($R_m \geq 40$) are considered unfeasible for engineering studies. Overall, more than 18,000 CPU core.days, or 50 CPU core.years, were utilized to deliver the presented results.

## 3. Results

The main purpose of this study is to assess the lower-nappe aeration (i.e., induced by the offset aerator) in the different model combinations. Hence, the results are presented and analyzed as follows. First, a global flow comparison. Second and most important, the air-water mixture and the lower-nappe aeration. Next, the maximum pressure increment at the spillway bottom, which is a frequent design criterion. Finally, a geometric parameter of the flow: the cavity zone length.

Globally, Figure 3 shows similar flow depth (solid blue surface) and velocity (dark-blue and red streamlines) in all model combinations. The only exception is the *CTPE* with *k–ε mixture* that, downstream of the impact zone, presents a significant increment of the water-depth and a smeared air-water interface. In all model combinations, the offset aerator separates the flow from the spillway bottom, creating a jet and a cavity zone filled with air immediately downstream of the aerator. At the cavity zone, the turbulence of the air-water lower interface of the jet promotes the entrainment of air into the flow. Thus, to feed this process, air enters continuously through the air-vents located laterally at the aerator.

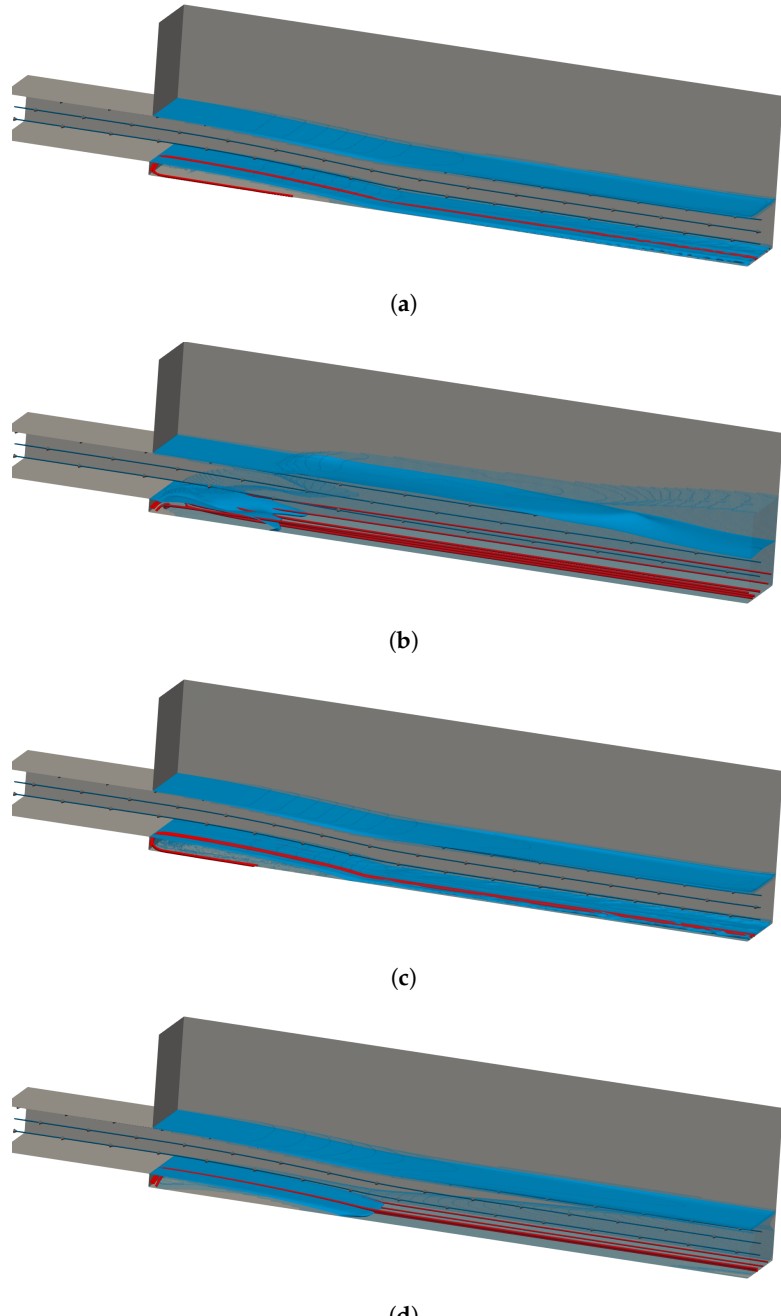

**Figure 3.** 3D flow—surface ($\alpha = 0.5$, solid blue), interface region ($0.01 < \alpha < 0.99$, transparent blue), air vents streamlines (red), water inlet streamlines (dark-blue) ($R_m = 40$, except $R_m = 30$ for *CTPE* $k–\epsilon$ *mixture*). (**a**) *TPVoF, k–$\epsilon$*. (**b**) *CTPE, k–$\epsilon$ mixture*. (**c**) *TPVoF, k–$\omega$ SST*. (**d**) *CTPE, k–$\omega$ SST Sato*.

### 3.1. Air-Water Mixture

Regarding aeration, the primary conditioning to the performance of the models is how they deal with the air-water interface and if the air is entrained into the water flow. Figure 4, shows the water fraction along the spillway, in an *xz* plane, for all model combinations and four mesh resolutions.

The major standout is the very distinct behavior of the *CTPE* with *k–$\epsilon$ mixture*. The lower-nappe aeration is much more intense, inducing an air-water mixing region at the bottom that occupies more than half of the flow depth. Moreover, it shows a significant air-entrainment at the upper water surface, resulting in a diffuse interface, which starts more upstream as the resolution increases. Therefore, the *CTPE* with *k–$\epsilon$ mixture* models

combination is not tested with the highest mesh resolutions: 40 and 60 cells per inlet height. These phenomena can be explained by the concept of the *k–ε mixture* turbulence model and are addressed in Section 3.5.

The second standout is the absence of an air-water mixture region at the bottom in the *TPVoF* combinations. Oppositely, this region is identified in the *CTPE*. Moreover, in the *TPVoF*, an air-water interface is present next to the spillway bottom, though becoming sharper as resolution increases. Thus, presenting a clear fluid separation and an airflow next to the bottom. This phenomenon is also noticed in the lower part of the water-fraction vertical profiles analyzed in Section 3.2.

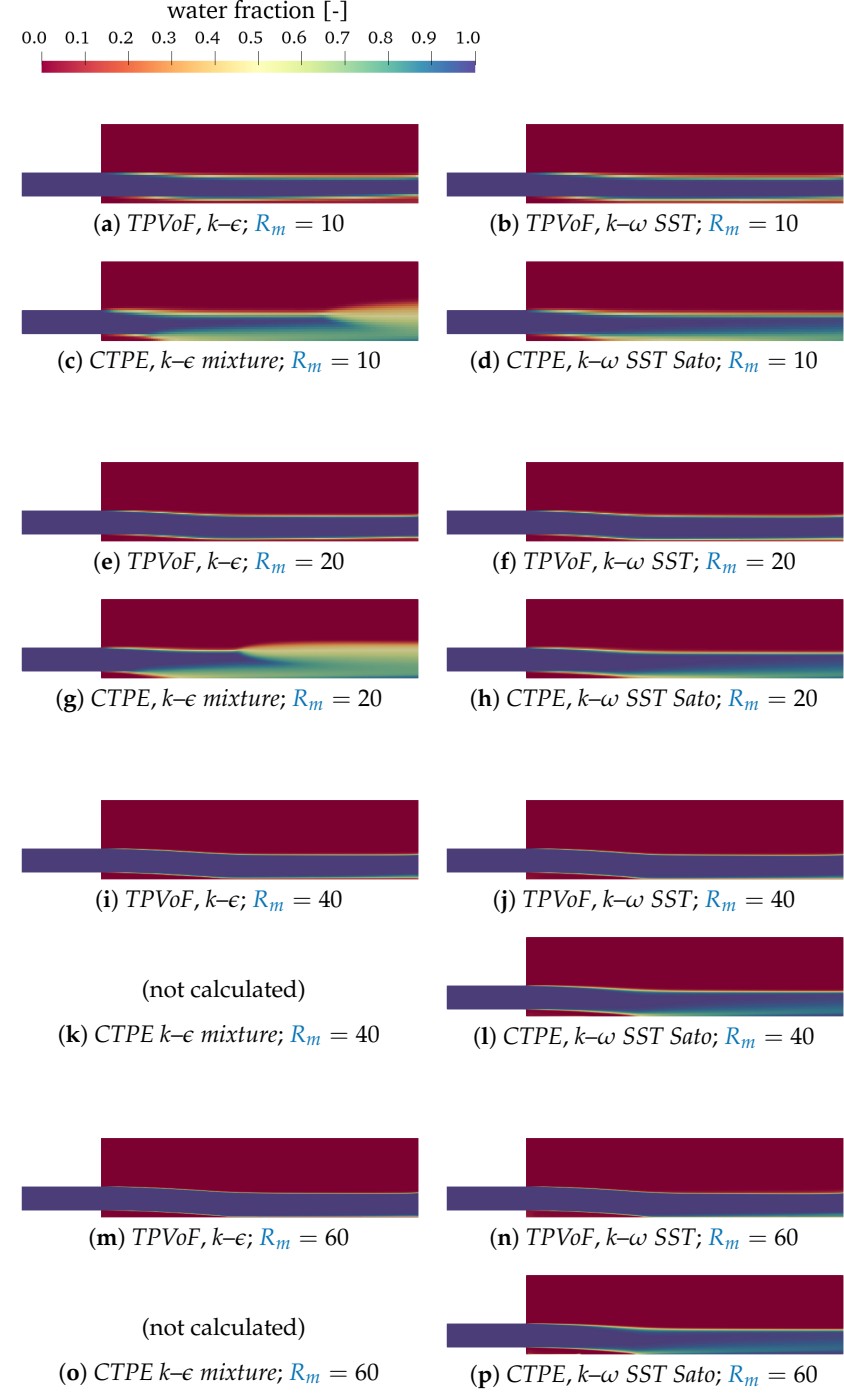

**Figure 4.** Water fraction ($\alpha$): vertical central plane.

### 3.2. Lower-Nappe Aeration

A spillway aerator is designed to entrain air into the water flow, eliminating cavitation damage next to the bottom and walls [5]. Air-concentrations ranging from 1 to 8% are needed to mitigate cavitation damage [8,69]. Further downstream of the aerator, air bubbles tend to rise to the flow surface, reducing the air-concentration next to the bottom. Therefore, it is frequent to find several aerators along a spillway.

Thus, the performance of a spillway aerator is determined by the amount of air entrained at the cavity zone (i.e., lower-nappe aeration) and in the air-concentration profile next to the bottom along the spillway.

Commonly, lower-nappe aeration is characterized by the ratio ($\beta$) between the flow-rate of air-entrained at the cavity zone ($Q_{al}$) and water flow-rate at the inlet ($Q_w$), see Equation (24). $Q_{al}$ is measured by the air flow-rate entering the cavity zone through both lateral air-vents (Figure 2).

$$\beta = Q_{al}/Q_w \tag{24}$$

Figure 5 shows the $\beta$ coefficient for all model combinations and different mesh resolutions. For low mesh resolutions ($R_m < 20$), all model combinations show the same behavior: $\beta$ is approximately twice the reference value ( $\beta^{ref} = 0.0475$ in [11]), decreasing as the resolution increases.

A significant result is that in the *TPVoF* lower-nappe aeration tends to a null value, constantly reducing with the increase of the mesh resolution.

The *CTPE* with *k–ω SST Sato* presents acceptable results for higher mesh resolutions ($R_m \geq 30$), converging to a lower-nappe aeration slightly inferior to the reference value. Oppositely, the *CTPE* with *k–ε mixture* reveals an unexpected increase of $\beta$ for $R_m \geq 20$, becoming even more abrupt for higher mesh resolutions. This behavior can be explained by the concept of the *k–ε mixture* turbulent model and is addressed in Section 3.5.

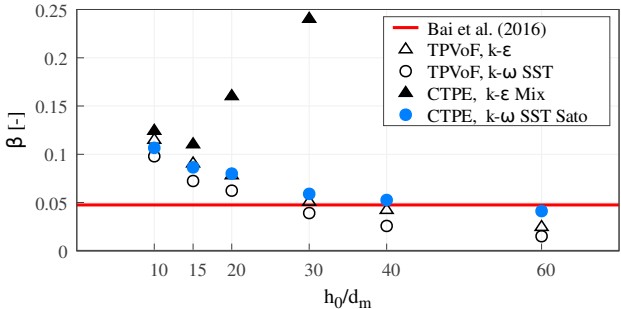

**Figure 5.** Lower-nappe aeration: ratio $\beta$ between the flow-rate of air-entrained at the cavity zone and water flow-rate, for distinct mesh resolutions.

The distribution of the air entrained due to lower-nappe aeration is analyzed in three vertical profiles of the water volume fraction ($\alpha$): at the cavity zone, at the impact zone, and downstream of the impact zone, i.e., 0.2, 0.7, and 1.0 m downstream of the step. Figure 6 presents these profiles for an intermediate mesh resolution ($R_m = 30$), which represents the behavior of the models.

For the three profiles, the *TPVoF* shows no air-water mixing. $\alpha$ is null at the bottom, and there is an abrupt transition to the complete water flow above, which is exacerbated with the mesh resolution increase. Thus, all the flow-rate of air from lower-nappe aeration is located next to the chute invert. The absence of air-water mixing is justified because the *TPVoF* model equations do not comprehend turbulent dispersion. Hence, any air-water mixing is due to the numerical diffusion in the interface, which is reduced with higher mesh resolution. To respect the *TPVoF* model conception, the mesh must be fine enough to reproduce the individual air-bubbles, which is unfeasible in engineering applications of spillways.

Better behavior is found in the *CTPE* with *k–ω SST Sato*. The mixed flow region is located in the lower half of the flow, sharing some similarities with the water fraction ($\alpha$) profiles of Bai et al. [11]. However, at the bottom, in the cavity zone $\alpha = 0.4$, and in the impact zone $\alpha = 0.7$, which is considerably inferior to the reference profile, where $\alpha$ is 0.8 and 0.92.

At the impact zone (see Figure 6b), the experimental data shows a mixed flow region with half the thickness of the flow and larger air-concentration in the inner part of the flow compared to the numerical models, except the *CTPE* with *k–ε mixture*. Downstream of the impact zone (i.e., 1.0 m downstream of the step, see Figure 6c)), despite the reference profile shows only a point at $z = 0.11$ m where $\alpha \approx 1$, the *CTPE* with *k–ω SST Sato* exhibits a thick layer ($0.08 \leq z \leq 0.13$ m) with $\alpha = 1$. The presented facts demonstrate the imperfect modeling of air-bubble vertical transport and turbulent dispersion, which may be justified by the single-size of the air-bubbles and the absence of significant roughness and oscillations in the air-water interface of the cavity zone due to the RANS framework, which is observed in Figure 1a).

The *CTPE* with *k–ε mixture* shows a low and non-realistic water concentration ($\alpha$) vertical profiles, exposing the exacerbated lower-nappe aeration and bubble diffusion. This behavior is associated with the *k–ε mixture* formulation and is explained in Section 3.5.

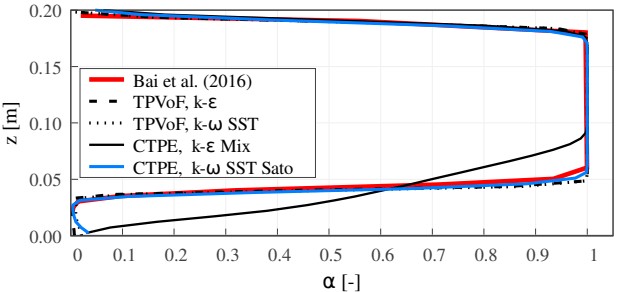

(**a**)

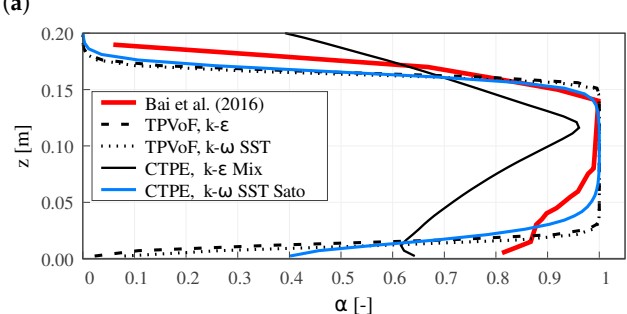

(**b**)

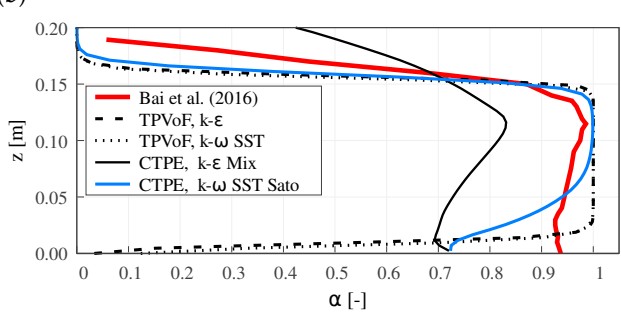

(**c**)

**Figure 6.** Water volume fraction vertical profile at $x = \{0.2, 0.7, 1.0\}$ m; $R_m = 30$. (**a**) $x = 0.2$ m. (**b**) $x = 0.7$ m. (**c**) $x = 1.0$ m.

### 3.3. Pressure Increment at the Spillway Bottom

The maximum pressure at the spillway bottom resultant from the jet impact is an important design criterion, especially in terms of materials lifetime. Thus, it is evaluated the pressure increment above the atmospheric pressure of 101325 $Pa = 1$ atm and compared against the laboratory data of Bai et al. [11]. The profiles of the pressure increment along the bottom ($\Delta p^b$) are presented in Figure 7. This analysis is focused on two properties: the value of the maximum pressure increment ($\Delta p^b_{max}$) and its location ($L_m$).

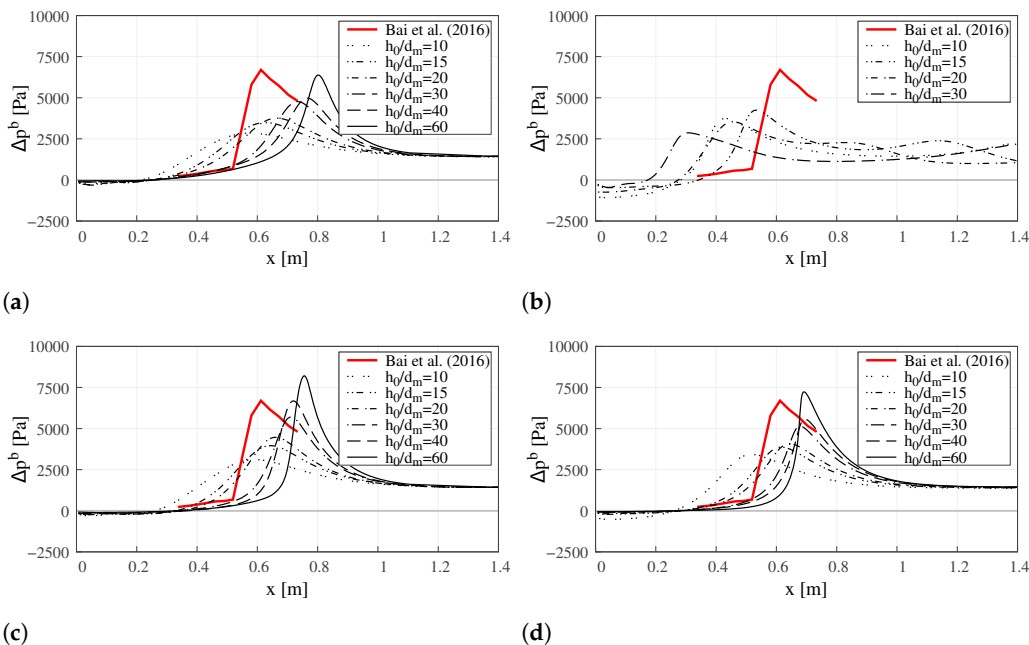

**Figure 7.** Pressure increment along the spillway bottom. (**a**) *TPVoF, k–ε*. (**b**) *CTPE, k–ε mixture*. (**c**) *TPVoF, k–ω SST*. (**d**) *CTPE, k–ω SST Sato*.

Regarding the maximum pressure increment ($\Delta p^b_{max}$, see Figure 8), both the *TPVoF* and *CTPE* models yield an increase of $\Delta p^b_{max}$ as the mesh resolution increases, which is even more noticed in the *k–ω SST* type models. Roughly, for the model combinations, the $\Delta p^b_{max}$ is only near to the reference value ($\Delta p^{b\ ref}_{max} = 6.7 \times 10^3$ Pa in [11]) for the higher mesh resolutions ($R_m \geq 40$). Although the *CTPE* with *k–ε mixture* shows simillar $\Delta p^b_{max}$ for the lower mesh resolutions ($R_m = \{10, 15\}$), at higher resolutions $\Delta p^b_{max}$ suffers an abrupt drop. For the different model combinations and the tested mesh resolutions, it is impossible to foresee a convergence of the $\Delta p^b_{max}$.

Besides the flow trajectory and momentum, the pressure next to the bottom is directly linked to local flow density, i.e., the mixture air-concentration. Thus, the results are coherent with the lower-nappe aeration (Figure 5) and the water volume-fraction profiles (Figure 6) presented in Section 3.2.

A sensitivity analysis of the mesh refinement near the walls was performed, evidencing that the results are not significantly affected. Hence, it is purposely not presented because it has a much smaller impact on the results than the remaining analyzed variables. Additionally to the mesh resolution near the bottom, the reproduction of the flow boundary layer, and the solver framework, including the pressure-velocity coupling scheme, may have a significant role in determining the pressure next to the bottom.

A y+ analysis was also performed, evidencing that the y+ condition for both models is never achieved (*k–ε*: 30 < y+ < 300, *k–ω SST*: 5 < y+ < 20). Moreover, an estimate based on the freestream flow (9 m/s) indicates that the height of the first cell next to the chute bottom should be approximately $h_0/1000$ to respect the *k–ε* y+ condition and even more to respect the *k–ω SST* y+ condition. Therefore, any significant resolution increase is incompatible

with practical engineering, even refining only near the walls. Nevertheless, the authors consider it is important to assess the turbulence models that are undoubtedly the most used with these solvers, in mesh resolutions representative of the commonly applied in research and practical engineering of hydraulic structures.

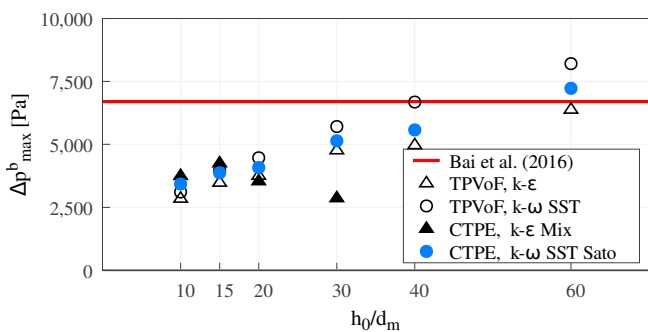

**Figure 8.** Maximum pressure increment at the spillway bottom.

The location of the maximum pressure increment $L_m$ is the distance along the chute bottom from the offset step to a point where the pressure increment is maximum ($\Delta p^b = \Delta p^b_{max}$), see Figure 9.

$L_m$ is similar for all model combinations, except for the *CTPE* with $k$–$\epsilon$ *mixture*, increasing in higher mesh resolutions. $L_m$ starts to converge in intermediate mesh resolutions $R_m \geq 30$ to a value approximately 15 to 35% larger than the reference ($L_m^{ref} = 0.62$ m in [11]). The *CTPE* with $k$–$\epsilon$ *mixture* shows abnormally small values for $R_m \geq 20$, due to the exacerbated jet diffusion. The *TPVoF* with $k$–$\epsilon$ and the *CTPE* with $k$–$\omega$ *SST Sato* present the peak of pressure nearer the reference value.

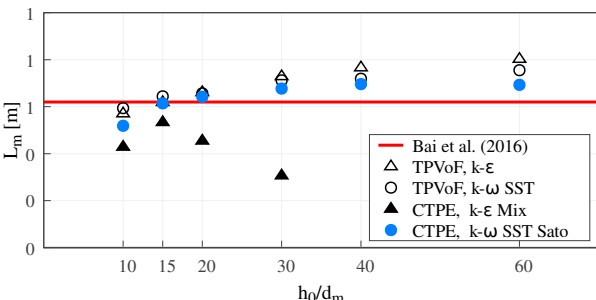

**Figure 9.** Maximum pressure increment location ($L_m$).

### 3.4. Cavity Zone Length

Following Bai et al. [11], the cavity zone length ($L$, see Figure 10) is the distance along the chute bottom between the offset step and the most upstream point of the impact zone where $\Delta p^b = 0.1 \times \Delta p^b_{max}$. $L$ also characterizes the jet trajectory.

In all model combinations using a $k$–$\omega$ *SST* type turbulence model, $L$ tends to converge to a value 15 to 20% larger than the reference ($L^{ref} = 0.52$ m in [11]). Comparing Figures 1a and 4, the air-water interface at the cavity zone is absent of significant roughness and oscillations in the air-water interface of the cavity zone due to the RANS framework. Also, the interface sharpness increases with higher resolutions. Thus, the lower interface of the jet is not diffuse enough, and the first impact point is located downstream than what is observed experimentally.

The *TPVoF* with $k$–$\epsilon$ seems to converge to a value nearer to the reference. The *CTPE* with $k$–$\epsilon$ *mixture* displays minimal values for $R_m = 30$, similarly to what is observed in $L_m$ (Figure 9).

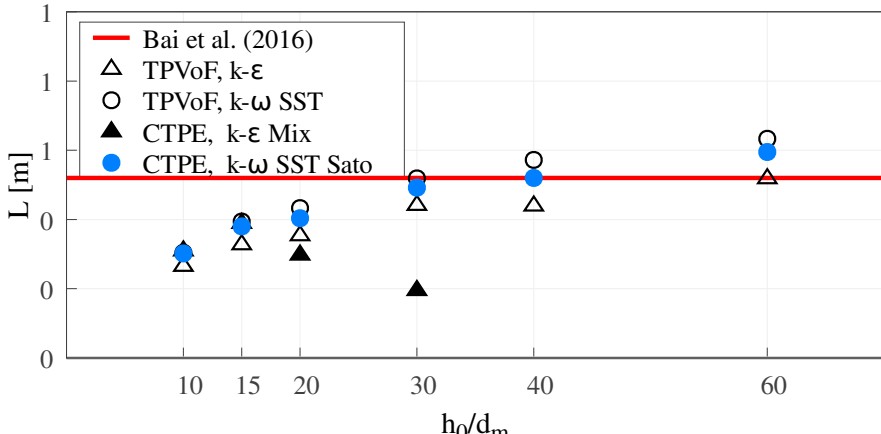

**Figure 10.** Cavity zone length (*L*).

*3.5. Turbulence*

A complete assessment of the turbulence is impossible due to the absence of detailed experimental data of $k$, $\epsilon$, $\omega$, and $\nu_t$ in [11]. Nevertheless, the four turbulence models are compared.

As presented in Figures 4 and 5, the $k$–$\epsilon$ and $k$–$\epsilon$ mixture present larger lower-nappe aeration than the corresponding $k$–$\omega$ SST and $k$–$\omega$ SST Sato. Analyzing the $k$ of the *TPVoF* in Figure 11 and the $k_w$ of the *CTPE* in Figure 12, the turbulent kinetic energy ($k$) is in the same order of magnitude at the air-water interface of the cavity zone. In detail, the *TPVoF* combinations show larger $k$ at the cavity zone, yet also have the smaller values of lower-nappe aeration. Despite the *CTPE* with $k$–$\epsilon$ mixture show the lower levels of $k_w$, it is the model combination with the highest amount of lower-nappe aeration. However, the turbulent viscosity ($\nu_t$) presented in Figures 13 and 14 expose the lower-nappe aeration is more significant where the $\nu_t$ is higher, which is displayed clearly by the $k$–$\epsilon$ and $k$–$\epsilon$ mixture models.

These results are justified by the conception of the models and their performance in the cavity zone, as reported by Bardina et al. [70]: "[the $k$–$\omega$ SST includes] a limitation of the growth of the eddy viscosity in rapidly strained flows. [...] The shear stress transport (SST) [...] improves the prediction of flows with strong adverse pressure gradients and separation.".

Additionally, in the *CTPE*, the turbulent dispersion is proportional to $\nu_t$ as defined in Equation (12). Oppositely, the *TPVoF* model equations do not comprehend the turbulent dispersion. Hence, any air-water mixing is due to the numerical diffusion in the interface, which is reduced with higher mesh resolution.

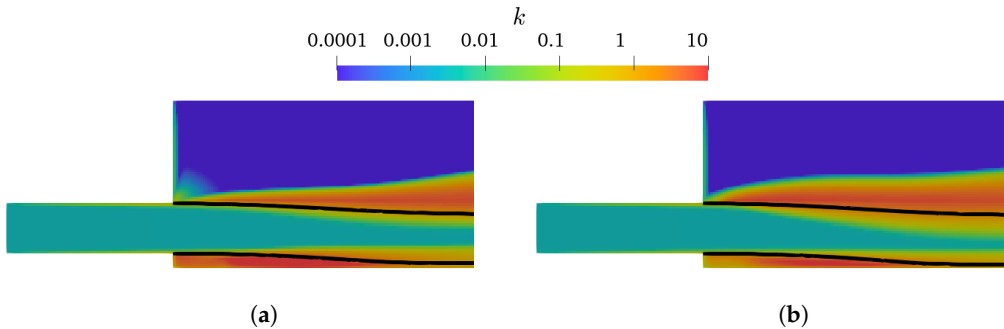

(**a**)             (**b**)

**Figure 11.** $k$ at vertical central plane of the *TPVoF* model with $R_m$ = 20; interface ($\alpha$ = 0.5, black line). (**a**) $k$–$\epsilon$. (**b**) $k$–$\omega$ SST.

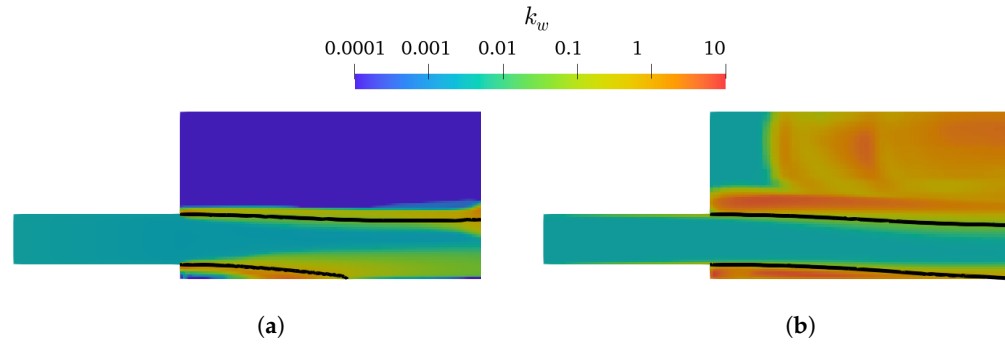

**Figure 12.** $k_w$ at vertical central plane of the *CTPE* model with $R_m$ = 20; interface ($\alpha$ = 0.5, black line). (**a**) *k–$\epsilon$ mixture*. (**b**) *k–$\omega$ SST Sato*.

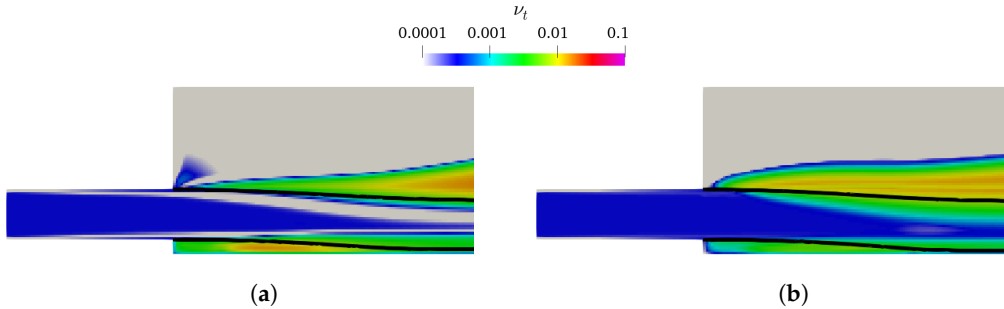

**Figure 13.** $\nu_t$ at vertical central plane of the *TPVoF* model with $R_m$ = 20; interface ($\alpha$ = 0.5, black line). (**a**) *k–$\epsilon$*. (**b**) *k–$\omega$ SST*.

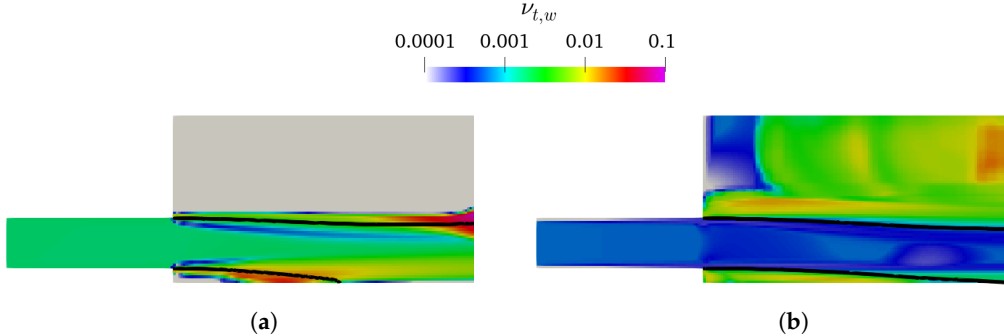

**Figure 14.** $\nu_{t,w}$ at vertical central plane of the *CTPE* model with $R_m$ = 20; interface ($\alpha$ = 0.5, black line). (**a**) *k–$\epsilon$ mixture*. (**b**) *k–$\omega$ SST Sato*.

The *k–$\epsilon$ mixture* model exacerbated lower-nappe aeration is explained by the conception of the model and the problems of the *k–$\epsilon$* formulation in rapidly strained flows and with large pressure gradients [70]. The *k–$\epsilon$ mixture* model solves a single set of equations for the air-water mixture, and the variables are calculated through an effective density-weighted average (19). Thus, this approach increases the gradients, especially at the air-water interface, which can be aggravated with the increase of the mesh resolution. This problem is not so significant in the *k–$\epsilon$*, because the implementation in the *TPVoF* does not include density explicitly, which, on the other hand, may lead to several turbulent-related issues at the fluids interface (i.e., spurious velocities and unrealistic turbulence production) [54,55]. Especially the high $k$ generation at the interface is observed in Figure 11. Moreover, the *k–$\epsilon$* type models exhibit minimal turbulent viscosity next to the chute bottom, which also conditions the inner turbulence fields.

### 3.6. Computational Cost

Although the same numerical schemes are employed for all model combinations, for stability reasons, the time-step definition required a maximum Courant number of 0.7 for the *TPVoF* and between 0.25 and 0.4 for the *CTPE* model. Nevertheless, the *CTPE* with

*k–ϵ mixture* for the higher mesh resolution is unstable for intermediate and high mesh resolutions. Nonetheless, an analysis is done considering the necessary CPU time to attain a stable solution during one second.

Globally, for the same model and mesh resolution, the calculation time is very similar between the two turbulent models. On average, the *CTPE* is 4 to 6 times slower than the *TPVoF*. The *CTPE* with *k–ϵ mixture* calculation time increased, especially due to spurious air velocities. On the other hand, for the mesh resolutions ($R_m$) of 10, 15, 20, 30, 40 and 60, there is an average increment of 4.5 times between two sequential resolutions. Hence, from $R_m$ = 10 to $R_m$ = 60, calculation time increases by a factor of $2 \times 10^3$. The *CTPE* with *k–ω SST Sato* shows very high levels of turbulence near the interface, which is probably due to the overestimation by two-fluid per-phase turbulence models of the turbulent kinetic energy near large scale interfaces with significant shear [71].

## 4. Discussion

The *TPVoF* and the *CTPE* Reynolds-averaged Navier–Stokes equations models are not directly comparable. The *TPVoF* is designed for two incompressible and immiscible fluids that share one set of momentum equations. Despite not including air-water mixing, the *TPVoF* is still used for engineering purposes. The *CTPE* is designed for two compressible and interpenetrating fluids, including heat transfer, and is more appropriate for highly-aerated flows. Besides the significant difference in the number and complexity of the mass, momentum, and energy conservation equations, the *CTPE* comprises several complex closure models (i.e., heat transfer, drag, lift, and turbulent dispersion). Thus, the calculation times and instability of the *CTPE* model are larger, and a single less accurate definition of the boundary conditions or the mesh may lead to pressure or temperature oscillations and solution divergence.

The interface modeling is also extremely different. The *volume-of-fluid* method is conceived to separate the fluids. Thus, the *TPVoF* does not comprise air and water mixing, and any air-water mixture present is due to numerical diffusion, especially in low mesh resolutions. In a RANS approach, with a mesh resolution compatible with most hydraulic engineering applications, the *VoF* model cannot reproduce the diversity of bubbles and air pockets present in the flow.

Oppositely, the *CTPE* allows the interpenetrability of both phases, which act as dispersed or continuous in each cell, according to the phase volume fraction. The mixing of the phases is mainly promoted by the turbulent dispersion, depending on the local turbulence conditions. However, the *CTPE* model has some limitations regarding bubbles and air pockets. First, the bubble size is limited to a single diameter, only varying due to pressure or temperature changes. Therefore, no bubble break-up or coalescence is considered. Secondly, the size of the bubbles must be at least an order of magnitude smaller than the size of the cells. Third, the interface tends to be more diffused than in the *TPVoF*, especially in the presence of high turbulence levels. Fourth, the RANS approach limits the growth of interface instabilities due to the limitations of the turbulence models and the respective spatial and temporal discretizations. The implementation of an expeditious sub-grid bubble model with a multiple-size bubble population in the *CTPE* could improve the range of spillway applications without significantly increasing the calculation time.

In the design of spillways, the *TPVoF* is the primary tool if aeration is nonexistent and in the absence of entrained air, benefiting from the easy model setup and low computational cost. The intake is the part of the spillway most suitable for the application of the *TPVoF*, considering that the flow is commonly in the subcritical regime, presenting low velocities and no air-entrainment. Downstream of the weir, where the flow changes from sub-critical to the supercritical regime, the *TPVoF* continues to be the best model until the spillway section where, due to the rise of the boundary layer, free-surface aeration initiates, or if in the presence of an aerator. Downstream, for the two analyzed solvers, the *CTPE* is the only tool that models the phenomena of aeration and air-bubbles transport and dispersion efficiently, despite showing some limitations due to the single-size bubble formulation. In some spillways, downstream of the chute, there is an energy dissipation basin where the

flow regime returns to subcritical, and the majority of the air-bubbles rise to the surface and are detrained. Hence, it may be useful to apply the *TPVoF* between the basin and the river restitution.

Despite not simulating the air-water mixture, the *TPVoF* may be an appropriate tool for preliminary analysis of flows with low air-concentrations (<20%), where the tracking of the free-surface is dominant. Further research should compare the efficiency of the *CTPE* with other successful approaches using the *VoF* or the mixture models combined with a sub-grid bubble model (e.g., [27,37]), in cases where the air-concentration is lower than 15–20%, or even for larger air concentrations, in light of some recent successful applications of the *VoF* and mixture models for highly aerated flows [16,18,24].

## 5. Conclusions

An assessment performed in a spillway offset aerator compares the *two-phase volume-of-fluid* (*TPVoF*) with the *complete two-phase Euler* (*CTPE*).

As expected, the *TPVoF* results depend highly on the mesh resolution, showing no air-water mixing. The accuracy for intermediate mesh resolutions is misleading: the lower-nappe aeration tends to null aeration as resolution increases. This is justified by the fact that the *TPVoF* model equations do not comprehend turbulent dispersion. Hence, any air-water mixing is due to the numerical diffusion in the interface, which is reduced with higher mesh resolution. To respect the *TPVoF* model conception, the mesh must be fine enough to reproduce the individual air-bubbles, which is inviable in engineering applications of spillways.

The *CTPE* combined with the $k$–$\omega$ *SST* turbulence model exhibits the most accurate results. Surprisingly, intermediate mesh resolutions are sufficient to surpass the *TPVoF* performance with reasonable calculation efforts, and no significant improvements are found in the highest resolutions, which demand exorbitant computational resources. Moreover, compressibility, flow bulking, and several entrained air effects in the flow are comprehended. Nevertheless, the turbulent dispersion of air-bubbles next to the bottom is not accurately reproduced, possibly due to the limitations of the single-size bubble population and the RANS approach to model the air-water interface at the cavity zone. Due to the more diffusive nature of the Euler-Euler approach, the *CTPE* may show slower convergence. Nevertheless, the lower-nappe aeration is expected to converge to a value similar to the highest resolution mesh. Contrarily to the *TPVoF*, the *CTPE* has a turbulent dispersion model which enhances the mixing of the phases.

The $k$–$\epsilon$ *mixture* turbulence model presents an exacerbated lower-nappe aeration, proving inadequate to simulate aeration in interfaces of rapidly strain flows and with high-pressure gradients.

Both the *TPVoF* and the *CTPE* show an increase of the maximum chute bottom pressure with higher mesh resolutions, surpassing the reference value, which is linked to the difficulties to mimic the air-concentration distribution next to the bottom.

Overall, despite not reproducing all aspects of the flow with acceptable accuracy, the *complete two-phase Euler* surpasses the *two-phase volume-of-fluid* model, evidencing an efficient cost-benefit performance and significant value in hydraulic engineering applications of spillway aerated flows. Further developments are expected to enhance the tool efficiency and stability. Nevertheless, the *two-phase volume-of-fluid* is appropriate to model the spillway intake and sometimes even the river restitution. Additionally, a comparison of the efficiency of the *CTPE* with other successful approaches using the *VoF* or the mixture models combined with a sub-grid bubble model is of major interest to identify the most appropriate model for specific hydraulic engineering applications of spillway aerated flows.

**Author Contributions:** Conceptualization, L.S.M. and J.L.L.; methodology, L.S.M., J.L.L. and M.T.V.; software, L.S.M.; validation, L.S.M., J.L.L. and M.T.V.; formal analysis, L.S.M.; investigation, L.S.M.; resources, L.S.M., J.L.L. and M.T.V.; data curation, L.S.M.; writing—original draft preparation, L.S.M.; writing—review and editing, L.S.M., J.L.L. and M.T.V.; visualization, L.S.M.; supervision, J.L.L. and M.T.V.; project administration, J.L.L.; funding acquisition, L.S.M. and M.T.V. All authors have read and agreed to the published version of the manuscript.

**Funding:** The authors acknowledge the following institutions for their funding and support: Fundação para a Ciência e a Tecnologia (FCT), Portugal—first author PhD Grant SFRh/BD/99815/2014; Laboratório Nacional de Engenharia Civil (LNEC), Portugal—first author PhD Grant co-funding.

**Institutional Review Board Statement:** Not applicable.

**Informed Consent Statement:** Not applicable.

**Data Availability Statement:** Not applicable.

**Acknowledgments:** The authors acknowledge the following institutions for their support: Laboratório Nacional de Engenharia Civil (LNEC), Portugal—first author PhD Grant and host; Instituto de Hidráulica Ambiental (IH Cantabria), Spain—first author host; Infraestrutura Nacional de Computação Distríbuida (INCD) funded by FCT and FEDER (European Regional Development Fund) under the project 01/SAICT/2016 nº 022153, Portugal—computational resources.

**Conflicts of Interest:** The authors declare no conflict of interest.

## Abbreviations

The following abbreviations are used in this manuscript:

| | |
|---|---|
| CFD | Computational fluid dynamics |
| *CTPE* | Complete two-phase Euler |
| DNS | Direct numerical simulations |
| DES | Detached eddy simulation |
| HPC | High-performance computing |
| LS | Level-set method |
| MX | Mixture model |
| RANS | Reynolds-averaged Navier–Stokes equations |
| SST | Shear stress transport |
| SGB | Sub-grid air-bubble density equation model |
| *TPVoF* | Two-phase Volume-of-fluid |
| *VoF* | Volume-of-fluid method |

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
