# Peer review of "Do the Volume-of-Fluid and the Two-Phase Euler Compete for Modeling a Spillway Aerator?"

_water, doi:10.3390/w13213092_

Round 1
Reviewer 1 Report
The paper presents a numerical study of a spillway aerator comparing several aspects such as mesh resolution, turbulence models, and air-water approaches (volume of fluid method VoF, and two-phase approach). The study analyses aspects such as the void fraction profile, the maximum pressure location and its value at the bottom of the spillway, and the characteristics of the cavity zone, comparing the numerical results with the published work carried out by Bai et al. (2016). The topic is within the scope of Water and the Special Issue.
The paper raises the importance of CFD calibration versus experimental modelling, and the difficulties of trying to model air-water flows, which I fully agree. However, there are some questions that should be considered by the authors (not in order of importance):
- After reading the paper, it is clear that none of the four configurations is able to reproduce the air-water flow behaviour more than some general aspects, and that the increment in the mesh resolution in some cases may drive to worse results without reaching a clear convergence to the experimental values. This should be indicated in a clearer way in the abstract and in the conclusions sections to avoid misunderstandings of the readers. The abstract should also indicate that the results were obtained with OpenFOAM as other codes may use different approaches and/or allow the use of additional limiting aspects in their multiphase models.
- The number of cited references is really large (up to 57 in the first and second pages, and 100 in the entire manuscript). This list should be largely reduced.
- The scheme in Fig.1b should show the same channel bottom angle before and after the aerator to be representative of the case study.
- In the Complete two-phase Euler model, which fluid was considered as a continuous phase, and which as a disperse phase? This may be quite different to other CFD codes in which both fluids may be considered as a continuous phase.
- Have the authors considered some mesh refinement near the walls? This may affect the results.
- Regarding the Numerical setup, please indicate how the turbulence for inflow was obtained (k = 3.1x10-5 m2s-2, epsilon = 1.6 x 10-6 m2s-3 and omega = 0.58 s-1).
- I am not sure if the numerical results were obtained in a single time step or considering the averaged value of a time window. Could the authors specify it? Is the model a transient or a steady state simulation?
- In the Fig. 3, what are the green surfaces? Is the isosurface obtained with a specific alpha value? In that case, the CTPE methods seem to have some problems near the walls. In addition, the d) configuration seems to solve a really large amount of air in the inlet channel, which should be zero, and zero air entrainment after the impact zone. Those values do not match with the large amount of air observed in Fig.1 after the impact zone.
- In the plots a, b, e, f of Fig. 4 it seems to be an 100% air layer near the bottom, that would not be physically representative of the experiment. In addition, other CFD codes that use VoF are able to model a minimal amount of air entraining into the water. However, Fig. 4 shows a sharp definition without air entrainment. Please revise if the setting of the models is limiting the model to have a more accurate representation of the phenomenon.
- Regarding Fig. 4, the only model that seems to reproduce the air entrainment in the water column observed in Fig. 1 is the CTPE k-epsilon. Please revise.
- In mostly of the plots, it is not specified if the data are obtained in a plane located in the center of the flume or in a plane located near one of the air vents. The values should be slightly different. Please indicate it.
- To analyse the behaviour of the numerical model, the paper should compare the chute in several cross-sections (e.g., at x = L/2, x = L, x = 2L…) to evaluate the availability of reproducing the air entrainment and the growing of the aerated layer.
- Fig. 8 may be affected by the lack of a mesh resolution near the bottom and/or the pressure/velocity coupling scheme. Please consider it.
- Considering the air layer observed near the bottom in Fig. 4, I do not know how the authors could obtain the cavity zone characteristics.
- Considering the large amount of available information to compare the numerical results with the experiment, I have doubts about including sections that cannot be compared / validated such as the Turbulence section. In my opinion, the study should be focused on the analysis of the free surface, air entrainment, and velocity field that may be compared versus the experimental campaign.
- Figs. 12b and 14b show what it seems a dependence of the boundary location.
Reviewer 2 Report
The authors compare to numerical concepts, the volume of fluid (VoF) and the Euler / two-fluid approach, for the numerical simulation of spillway aerators. The manuscript addresses a topic which is of interest for the hydraulic engineering community and thus suited for the readership of Water. The paper is well written, and the line of thought is easy to follow. The representation of results is OK but can be slightly improves (see minor comments below). However, I have some major concerns about the methodology and think that the interpretation of results may needs to be revised in some points. I structured my comments into major and minor:
Major
- General: the authors benchmark the CTPE approach against a VoF solver. This is somewhat confusing as the VoF approach is clearly not able to represent the phenomena of air entrainment (at least not with the grid and setup of this study). This makes the main conclusion of the paper – that CTPE is better suited than VoF – superficial. Arguing that VoF is the most used method in hydraulic structures falls short, as any CFD expert would not apply it to aerated flows. Why did the authors not compare the CTPE approach to a method that can reproduce air entrainment (e.g. mixture models)?
- Introduction, page 3: the authors claim that the CTPE is somehow more physically sound than single fluid approaches with sub-grid bubble and air entrainment models. This is not true per se. All momentum transfer models in the CTPE context are empirical (e.g. drag, lift, and especially dispersion) and thus do not overcome the limitations of other empirical sub-grid models. In this context, the authors fail to cite successful applications of sub-grid air entrainment models to aerated flows in hydraulic structures (e.g. Hohermuth et al. https://doi.org/10.1080/00221686.2020.1780492)
- Mathematical models, page 7: the turbulent dispersion force is the model which is mainly responsible for the simulation of air entrainment in this context. Therefore, it needs to be introduced in more detail, what are the underlying assumptions? Is the model actually made for this situation? In which region is the model activated? (at least in older versions, the dispersion model was only active in regions with phase fractions below 0.3)
The concept of phase fractions and blending between “air dispersed in water” and “water dispersed in air” needs to be introduced somewhere. - Mathematical models, page 5: the employed version of interFoam does not consider density in the turbulence models. This is physically unreasonable and leads to incorrect results near interfaces with strong density differences and significant shear. Please read and cite the paper of Fan & Anglart (https://doi.org/10.1016/j.cpc.2019.106876) in this context. This point should also be discussed when showing turbulence quantities in section 3.5.
- Numerical setup, page 8: It seems like the same computational grids were use for the k-e and the k-omega SST models. Depending on the implementation, k-omega SST models benefit (some actually require) enhanced near-wall mesh resolution. Did the authors test if the SST models in their study were sensitive to wall-refinement? What were typical y+ values for the different grids?
- Section 3.5 on turbulence: the authors seem unaware of general limitations of CTPE approaches in stratified flows. It is known, that (at least k-e based) two-fluid per-phase turbulence models tend to overestimate turbulent kinetic energy near large scale interfaces with significant shear. In this context, please read and cite Dong et al. (https://doi.org/10.1016/j.ces.2021.116975) and relevant literature cited therein. How did this limitation affect the authors results?
- Section 3.5 on turbulence: the k-e mixture model also accounts for bubble induced turbulence in regions with low void fraction. To what extend did this contribute to the observed behavior of the TPVoF model in this study? Turbulence modelling in situations with air entrainment is important and the discussion in this section should be improved.
- Discussion, page 19: CTPE is not the only model approach that can model air entrainment phenomena. The authors should properly acknowledge the performance achieved with mixture model approaches (see e.g. Valero & Garcia-Bartual https://doi.org/10.1007/978-981-287-615-7_38)
- Conclusion:
- the statement that TPVoF results depend highly on mesh resolution is not supported by the presented data. The solver shows similar convergence as the CTPE approach in all Figures. Possibly, this comes from the idea that TPVoF should be able to reproduce entrainment, which it clearly cannot. Also, wouldn’t one actually expect CTPE to show slower convergence due to the more diffusive nature of the Euler-Euler approach?
- The statement that CTPE surpasses TPVoF performance is of limited use as TPVoF is not intended to reproduce the phenomena at hand. Comparing CTPE to a VoF solver with sub-grid air entrainment models would make much more sense.
Minor
- What is the green color in Fig. 3? Why does there seem to be an interface region in the inlet section of CTPE with k-w SST Sato model?
- Why is the CTPE at 0.7 at the bottom in Fig. 6? This needs special attention given that 0.7 is treated as the lower end of “water dispersed in air” by the empirical momentum transfer models in twoPhaseEuler.
- Section 3.4: why is the cavity zone length significantly larger in all models? Please comment on this.
- Discussion, page 19 “Third, the interface tends to be blurrier than in the TPVoF, especially in the presence of high turbulence, which makes the calculation of the surface tension in the interface of large air-pockets more difficult.” This statement seems out of context. None of the tested approaches reconstructs the interface accurate enough to derive surface tension.
Given the above-mentioned major issues, I recommend a major revision of the paper and urge the authors to identify and discuss the limitations of the employed model approaches more carefully. In my opinion, the main conclusion that CTPE is the best method to simulate spillway aerators is not supported by the current results and discussion. I acknowledge, that the authors performed a significant effort in terms of computational workload, and I would be happy to look at a revised version of this manuscript in the near future.
Reviewer 3 Report
The submitted work compares the efficiency of the two-phase volume of-fluid (TPVoF) with the complete two-phase Euler (CTPE). A spillway with an offset aerator device is used to evaluate both models’ compliance in different 3D mesh resolutions and turbulence models. The TPVoF and the CTPE models are evaluated against the physical modeling of an offset chute aerator by Bai et al. (2016). For the execution of the numerical simulations, the interFoam and twoPhaseEulerFoam solvers included in thethe OpenFOAM® toolbox has been used.
The topic is of interest and the proposed objective is ambitious, since different approaches and turbulent closure models are considered.
The Introduction section is clear and rather complete. The problem at hand is well stated.
Some papers are reported as a simple list (e.g. [12-26], [27-44]), try to reduce.
I suggest to include the following paper:
- Lauria A, Alfonsi G (2020). Numerical investigation of ski jump hydraulics. JOURNAL OF HYDRAULIC ENGINEERING, vol. 146, 04020012, ISSN: 0733-9429, doi: 10.1061/(ASCE)HY.1943-7900.0001718
Also the section Methodology seems to be clear and rather complete. Add some details about the decomposition method used for MPI.
The Results and Discussion sections seems to be clear, tables and figures are easy to interpret showing essential data.
Round 2
Reviewer 1 Report
I appreciate the great effort carried out by the authors to solve the previous questions/doubts of the reviewers due the time constraints of the journal.
Mostly of my questions in this step are related with the previous one, which should be that should be clarified a bit more (not in order of importance):
- As the models allow (in a better or in a more limited way) to reproduce the flow trajectory and the pressures on the stagnation point, but not the air entrainment, some comments about the limitation of the air-water approached to model the void fraction (e.g., 0.1-0.2) would be appreciate, considering that the goal is the free surface definition, not modeling the void fraction experimental results in the entire water column.
- Regarding the pressure study, I would like to know if the authors analyzed the setting of the pressure/velocity coupling scheme (e.g., Rhie Chow order).
- The mesh refinement near the bottom may depend on the CFD code used. Some of them allow to solve meshes with special refinement near the walls, such as inflation layers, that allow to reach the y+ value required by each turbulence model. In addition, some codes allow to include wall functions to the turbulence models. Their use may improve the results located near the boundary layer. Have the authors analyzed if OpenFOAM may offer those possibilities?
- Regarding the data sampled and averaged, please indicate the time windows and the amount of data for the averaging (e.g., 20 frames per second during 10 seconds).
- As I previously suggested, I would prefer to find more numerical results versus the Bai et at. (2016) experimental conditions, instead of the “3.5 Turbulence section”. In that way, the analysis of Fig. 6 may be extended to further cross sections located in the cavity, impact, and equilibrium zones, as Bai et at. (2016) measured 14 cross sections between 0.19 < x/L < 1.98. This type of comparisons would be highly appreciated by the readers of the paper.
Reviewer 2 Report
The authors have significantly revised the manuscript and provided detailed replies to all my questions and comments. In general, my comments were addressed satisfactorily and I can follow the authors’ argument on going a comparison based on “typical practice in the field of hydraulic engineering”. However, I want to emphasis that Euler-Euler models have been standard engineering tools in other disciplines such as nuclear and chemical engineering. The main remaining issue where I tend to disagree with the authors is the applicability of mixture models. The authors state that mixture models can only be applied up to 15-20% void fraction, citing Bombardelli 2012. Unfortunately, I cannot access the conference contribution from Bombardelli online to follow his reasoning. The seminal textbook on two-phase CFD by Ishii and Hibiki state the following limitation for mixture (drift-flux) models:
“It is generally accepted that the drift-flux model is appropriate to the mixture where the dynamics of two components are closely coupled. This suggests that the same argument may be used for the macroscopic two-phase flows. The usefulness of the drift-flux model in many practical engineering systems comes from the fact that even two-phase mixtures that are weakly coupled locally can be considered, because the relatively large axial dimension of the systems usually gives sufficient interaction times.”
As drift velocities between air and water in hydraulic engineering applications are typically small, I think that strong coupling is an assumption with reasonable accuracy in most cases. Also note that e.g. Valero & Garcia-Bartual 2014 and Hohermuth et al. 2021 successfully applied mixture models to a wider range of void fractions in a hydraulic engineering context.
A further minor comment is related to the color schemes; Fig. 4 looks very nice, but the authors use different colormaps in Figs. 11 – 14. I suggest to use the same colormap (from Fig. 4) throughout the manuscript.
Overall, I think the manuscript is suitable for publication after addressing the comments above. Since these comments are minor, I do not need to see the manuscript again before publication.
